# Predictive learning shapes the representational geometry of the human brain

Antonino Greco [1,2,3] ✉, Julia Moser [4,5], Hubert Preissl [4,6,7,8,9] &
Markus Siegel [1,2,3,6] ✉

Predictive coding theories propose that the brain constantly updates internal models to minimize prediction errors and optimize sensory processing. However, the neural mechanisms that link prediction error encoding and optimization of sensory representations remain unclear. Here, we provide evidence how predictive learning shapes the representational geometry of the human brain. We recorded magnetoencephalography (MEG) in humans listening to acoustic sequences with different levels of regularity. We found that the brain aligns its representational geometry to match the statistical structure of the sensory inputs, by clustering temporally contiguous and predictable stimuli. Crucially, the magnitude of this representational shift correlates with the synergistic encoding of prediction errors in a network of high-level and sensory areas. Our findings suggest that, in response to the statistical regularities of the environment, large-scale neural interactions engaged in predictive processing modulate the representational content of sensory areas to enhance sensory processing.

Living organisms must adapt to ever-changing complex environments. To accomplish this, it is advantageous to anticipate environmental changes that posit threats or opportunities to survival. For this reason, the brain is able to detect and extract statistical regularities in sensory inputs, an ability that has been referred to as statistical learning and which humans are especially capable of[1–3].

Predictive coding theory provides a framework to explain how regularities are extracted from sensory inputs and how they are used to optimally predict future outcomes[4–6]. This framework generally assumes that the brain possesses a generative internal model of the environment (latent variables that cause the sensory observations), and updates the model by computing the prediction error between its probabilistic predictions and the sensory data[7–11]. These predictive

mechanisms are thought to give rise to increased neural activity following an unexpected sensory input[12,13] or decreased neural responses when input is expected[14,15].

In the auditory domain, the oddball paradigm has been widely adopted to probe the ability of the brain to track statistical regularities[16]. A rare deviant tone presented within a sequence of regular tones elicits an event-related response, the so called mismatch negativity (MMN)[17]. Under the predictive coding framework, this response is interpreted as a neural signature of the prediction error between the expectation of the generative model (regular tone) and the sensory data (deviant tone)[18,19]. Furthermore, recent studies on auditory statistical learning that used more complex patterns of sound sequences showed that cortical responses do not only reflect

[1]Department of Neural Dynamics and Magnetoencephalography, Hertie Institute for Clinical Brain Research, University of Tübingen, Tübingen, Germany. [2]Centre for Integrative Neuroscience, University of Tübingen, Tübingen, Germany. [3]MEG Center, University of Tübingen, Tübingen, Germany. [4]IDM/fMEG Center of the Helmholtz Center Munich, University of Tübingen, Tübingen, Germany. [5]Masonic Institute for the Developing Brain (MIDB), University of Minnesota, Minneapolis, USA. [6]German Center for Mental Health (DZPG), Tübingen, Germany. [7]German Center for Diabetes Research (DZD), Tübingen, Germany. [8]Department of Internal Medicine IV, University Hospital of Tübingen, Tübingen, Germany. [9]Department of Pharmacy and Biochemistry, University of Tübingen, Tübingen, Germany. ✉e-mail: antonino.greco@uni-tuebingen.de; markus.siegel@uni-tuebingen.de

violations of sensory predictions at a local tone level but also on the global sequence level[20–25].

In sum, a large body of evidence has provided insights into cortical signals compatible with the encoding of prediction errors. In contrast, little is known about how such signals are used to update the brains internal generative model. Studies on perceptual learning suggest the plasticity of sensory representations, even in low-level sensory regions, to optimise sensory processing[26–28]. Furthermore, studies on statistical learning show that the similarity of neuronal representations of sensory stimuli, i.e. of neural activity patterns that are specific for distinct stimuli[29], reflects the learned statistical dependencies between these stimuli[30–32]. This suggests that statistical learning may shape the geometry, i.e. mutual similarities, of neural representations to match the geometry, i.e. mutual statistical dependencies, of sensory inputs. However, the neural mechanisms underlying this learning remain unclear. If prediction error signals are used to update neuronal representations, these two phenomena should be linked, i.e. the neuronal encoding of prediction errors should be correlated with the updating of neuronal representations. However, so far evidence to support this fundamental link between prediction errors and the updating of neural representations is missing. Here, we sought to establish this link.

We performed magnetoencephalography (MEG) recordings in human participants passively listening to acoustic tone triplet sequences with low or high regularity. Representational similarity analysis[33] revealed that, over the course of the experiment, the brain aligned its representational geometry to match the statistical structure of the sequences. We defined predictive learning as the process of minimising prediction errors about future sensory observations and employed computational modelling to derive neural signals encoding prediction error trajectories[34–36]. We found that the strength of prediction error encoding indeed predicted the magnitude of the alignment of sensory representations. Furthermore, based on Partial Information Decomposition[37], we found that brain regions that showed representational alignment also engaged in a synergistic encoding of prediction errors. Our findings suggest that in the human brain, large-scale neural interactions engaged in predictive processing modulate the sensory representational geometry in response to the statistical regularities of the environment.

## Results

We recorded MEG in 24 human participants who passively listened to two sequences of 12 acoustic tones[38]. Each tone had a duration of 300 ms followed by a 33 ms silent gap. For the sequence construction, tones were grouped into triplets (1 s triplet duration) where the tones in a triplet never spanned more than one octave. Both sequences consisted of a total of 800 triplets. One sequence, which we are referring to as the high regularity (HR) condition, consisted of only four different types of triplets, while in the other sequence, which we are referring to as the low regularity (LR) condition, the order of tones inside a triplet was changing throughout the sequence (Fig. 1a). The different regularities of the two sequences are reflected in their distinct transition matrices between consecutive tones (Fig. 1a right).

A previous behavioural analysis of the dataset showed that participants learned the statistical regularities of the tone sequences[38]. After listening to both stimulus sequences, participants rated triplets from the high regularity sequence as significantly more familiar than partial or random triplets. Thus, participants showed significant statistical learning through passive stimulus exposure.

### Representational geometry aligns with sensory input statistics

To investigate this learning effect at the neural level, we source-reconstructed neural activity throughout the brain from the MEG data using the Desikan-Killiany parcellation scheme[39] and beamforming[40]. As expected, both tone sequences evoked neural responses that peaked about 60 ms after each tone onset in bilateral auditory cortices (Fig. 1b) and at about 110 ms in the left temporal cortex (Supplementary Fig. S1). Thus, source reconstruction yielded robust cortical responses. We next applied multivariate decoding and representational similarity analysis (RSA)[33] on the source-level brain activity to investigate the neural representation of different tones and to test if this representation changed during learning (Fig. 2). We quantified the distance between neural population responses to different tones using the cross-validated Mahalanobis distance (cvMD)[41,42] (Fig. 2a). This yielded representational dissimilarity matrices (RDMs) that quantified the distance of neural representations for all pairs of tones. Importantly, we performed this analysis temporally resolved relative to each tone presentation and separately in five consecutive blocks of trials throughout each sequence. This allowed us to resolve the temporal dynamics of neuronal tone representation on a fast timescale in response to each tone and on a slow timescale throughout learning. As

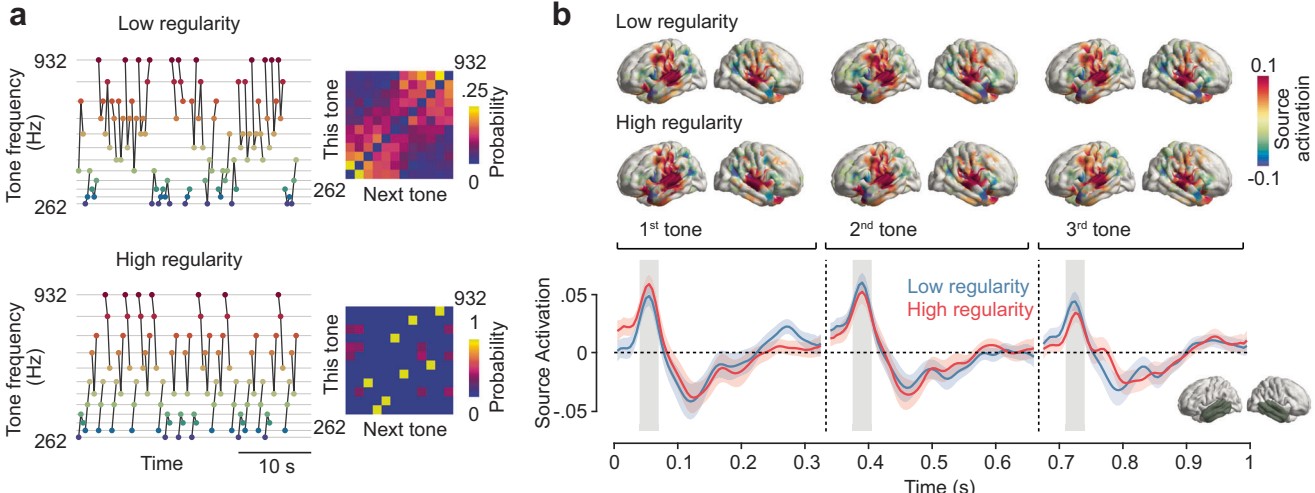

**Fig. 1 | I Experimental design and auditory cortical responses.** Subjects passively listened to two sequences of 12 acoustic tones. **a** left: exemplary sections of low and high regularity sequences arranged by their frequency. Grey lines indicate triplets. Right: transition matrices of subsequent tones for both sequences. **b** top: cortical distribution of evoked activity 50–70 ms post onset of each of the three tones in a triplet. Bottom: source-reconstructed evoked activity in bilateral temporal cortices (bottom right inset) across triplets in the low and high regularity condition. Shaded areas indicate the standard error of the mean (SEM).

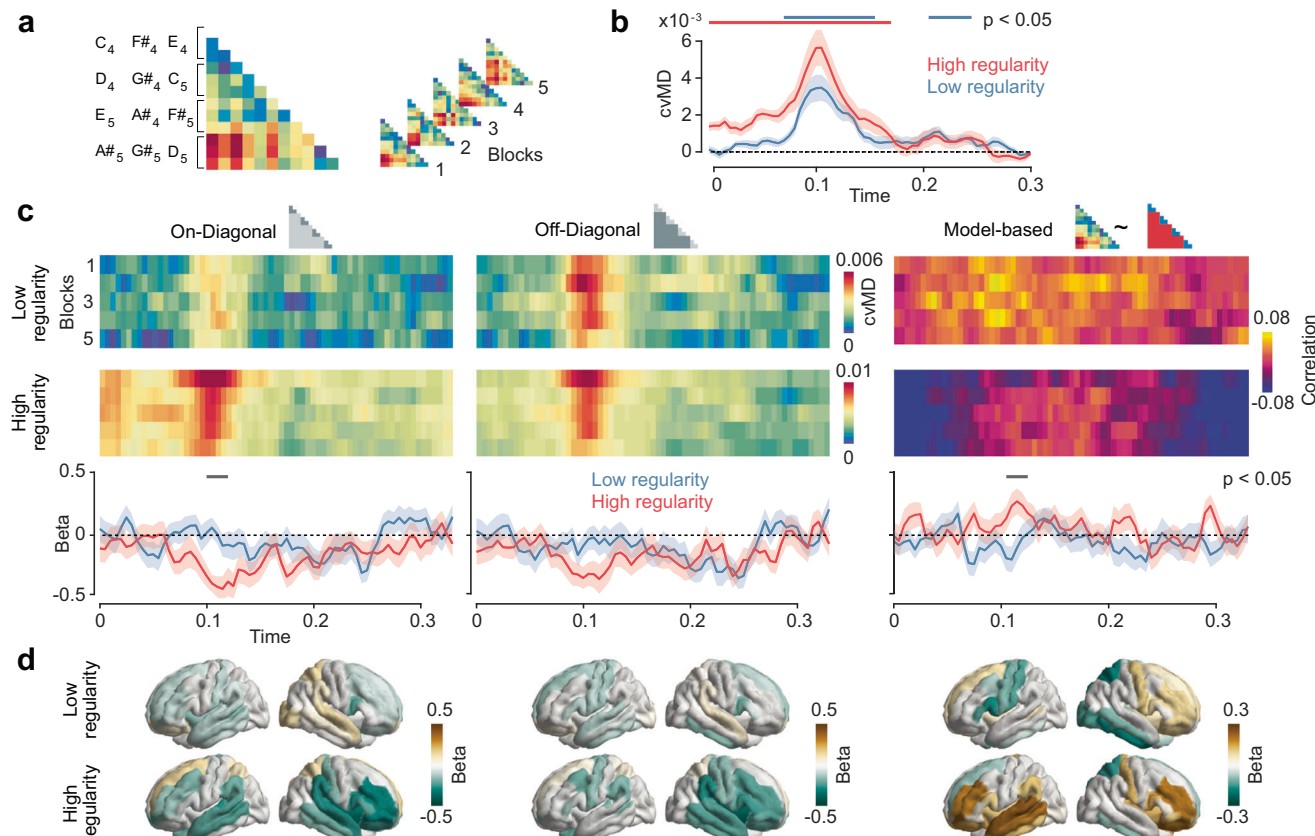

**Fig. 2 | I Representational similarity analysis (RSA) reveals a representational shift for the high regularity sequence. a** Analysis pipeline. Representational dissimilarity matrices (RDMs) were computed based on the cross-validated Mahalanobis Distance (cvMD) between all pairs of tones and ordered according to the triplet structure in the high regularity sequence. RDMs were computed for 5 consecutive blocks of trials. **b** Time course of the average RDM for tones in both sequences. Shaded areas indicate SEM, and horizontal lines indicate statistical significance ($p < 0.05$), which was determined by cluster-based paired two-tailed permutation tests. **c** Top: on- and off-diagonal cvMD and model-based RSA (Spearman correlation) plotted as a function of time and blocks. Bottom: Regression slopes of cvMD values and RSA coefficients across blocks. Shaded areas indicate SEM, and horizontal lines indicate statistical significance ($p < 0.05$), which was determined by cluster-based paired two-tailed permutation tests. **d** Searchlight RSA for on- and off-diagonal cvMD values and model-based estimates across the brain in the 110–120 ms time window. Significance was determined by cluster-based paired two-tailed permutation tests.

a first step, we averaged all RDMs across tone pairs and blocks. We found that tones were well decodable for both sequences, with peak decoding performance around 100 ms post tone onset (Fig. 2b, $p < 0.0001$ cluster-corrected; peak Cohen's $d = 1.46$).

We next ordered RDM entries such that values near and off the diagonal represented representational distances of tones within and between triplets in the HR sequence, respectively. Then, we separately quantified neural distances of tones within (Fig. 2c left) and between triplets (Fig. 2c middle) across time and blocks. To quantify the representational dynamics during learning, we computed the linear slope of the average representational distance within and between triplets across sequence blocks (Fig. 2c bottom). We predicted that if learning matched the representational geometry of neural representations to the statistical regularity of sequences, then distances of tones within triplets should decrease across blocks more than of tones between triplets, and this effect should be specific for the high regularity condition. Indeed, this is what we found.

For the high regularity condition, around 120 ms post tone onset, representational distances within triplets decreased across blocks (negative slope), and this decrease was significantly stronger for the high as compared to low regularity condition (Fig. 2c left; $p = 0.008$ cluster-corrected; $d = 0.79$). There was also a trend for representational distances to decrease between triplets, but there was no significant difference in slopes between conditions (Fig. 2c right; $p > 0.05$

cluster-corrected). To directly test our prediction, we then performed a model-based RSA and fitted a theoretical RDM in which the within-triplet distances were lower than between-triplet distances (Fig. 2c). As predicted, around 120 ms the model-fit increased across blocks in the high regularity condition, and this increase was significantly stronger than in the low regularity condition ($p = 0.038$ cluster-corrected, $d = 0.84$).

Which brain regions showed this updating of sensory representations? To address this question, we repeated the analysis in a searchlight fashion across the cortical surface for the time interval from 110 ms to 120 ms post-tone onset (Fig. 2d). We found that the decrease of representational distances across blocks within and between triples was strongest in dorsolateral prefrontal and temporal regions (Fig. 2d). Also, the model-based RSA showed that the increase of the model fit across blocks, i.e. the relative decrease of representational distances within triplets, in the high regularity condition peaked in bilateral dorsolateral prefrontal cortices and in left temporal cortex (Fig. 2d). In sum, these findings showed that the brain changed its sensory representations to adaptively match the statistical structure of the sensory inputs. Specifically, in the context of predictable tone triplets, the brain updated the tones' representations in a way that made them more similar between tones belonging to the same triplet as compared to tones belonging to different triplets.

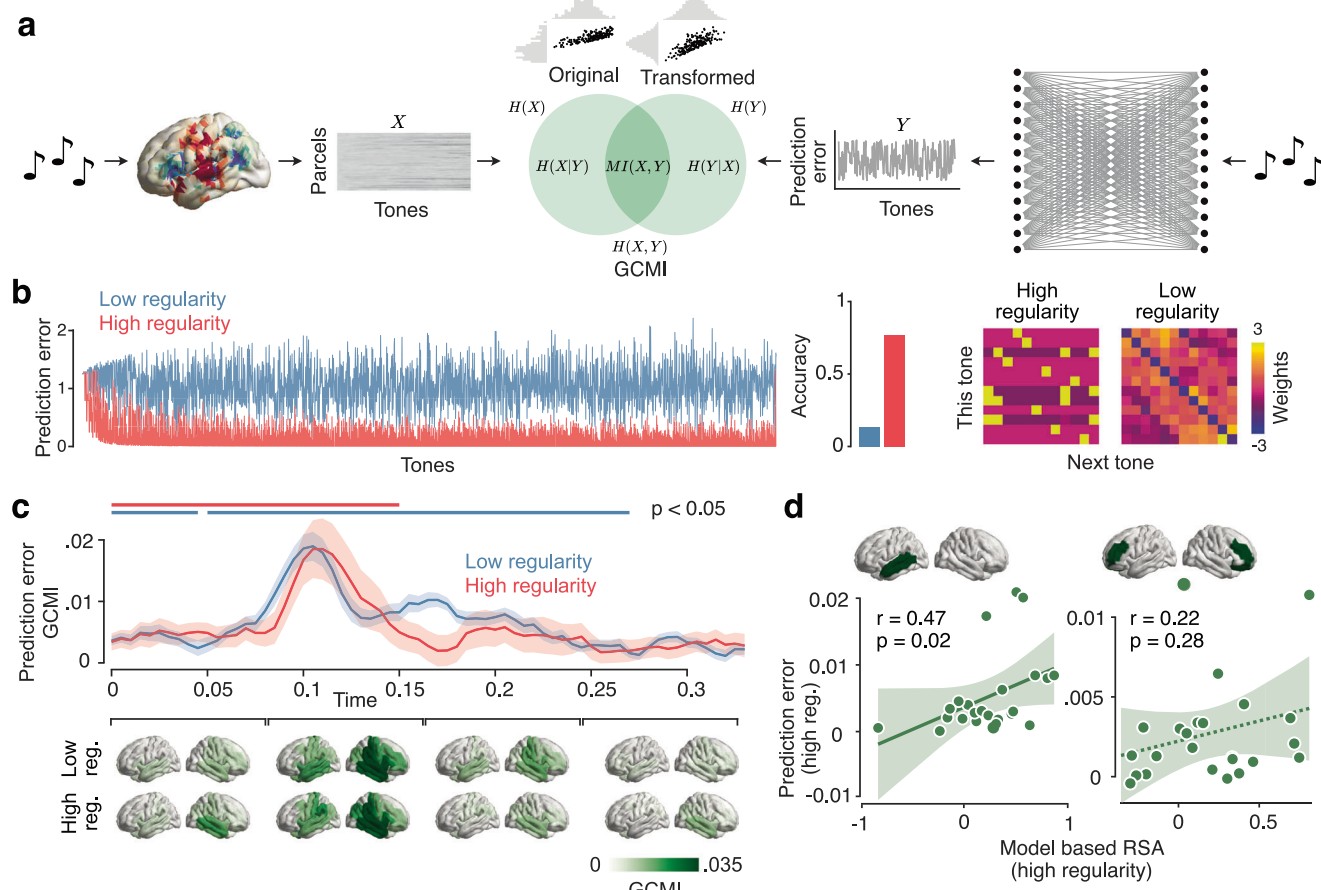

**Fig. 3 | I Computational modelling of prediction error trajectories using an ideal observer model. a** analysis framework for fitting prediction error trajectories from the model to each parcel of the brain. Left: For each time point relative to a tone presentation, the brain data is structured as a matrix with parcels and tones. Right: prediction error trajectory extracted from the ideal observer model. Centre: illustration of Gaussian Copula Mutual Information (GCMI). **b** Left: prediction error trajectories extracted from the model for both sequences. Middle: prediction accuracies of the model. Right: weight matrices of the ideal observer model after training on the full sequences. **c** top: Time-course of prediction error encoding (GCMI) for both sequences. Shaded areas indicate SEM, and horizontal lines

indicate statistical significance ($p < 0.05$). Significance was determined by cluster-based paired two-tailed permutation tests. Bottom: cortical distribution of prediction error encoding in 4 time windows. **d** correlation analysis between the representational shift and the prediction error encoding for the brain regions indicated on the top. Dots represent individual participants. Shaded areas indicate 95% confidence intervals. The dotted lines indicate non-significant regression fits. Solid lines indicate significant regression fits. Significance ($p < 0.05$) was determined using the Pearson correlation coefficient and Bonferroni correction. Source data are provided as a Source Data file.

## Large-scale encoding of prediction error

After establishing how statistical learning updated sensory representations, we next focussed on the neural encoding of errors between sensory inputs and input predictions[6,34]. We adopted a computational modelling approach to examine how the brain encoded the prediction error. We used an ideal observer model (Fig. 3a), a perceptron neural network resembling the Rescorla-Wagner model for categorical data[34], that predicted the next tone given the previous one. Inspired by Bayesian models of predictive coding, the model employed a dynamic learning rate that changed as a function of the ideal observer's uncertainty[43]. After fitting the model to each stimulus sequence, the weight matrix captured the actual transition matrix of each sequence (Fig. 3b, compare Fig. 1c). Also, the prediction error and accuracy of the fitted models reflected the statistics of the two sequences with lower prediction error and higher accuracy for the more predictable high regularity condition (Fig. 3b).

We then extracted the prediction error trajectories of the model for each condition (Fig. 3b) and tested if prediction errors were encoded by the neural activity using Gaussian Copula Mutual Information (GCMI)[44] (Fig. 3a). We found that the prediction errors were indeed significantly encoded, peaking around 100 ms post tone

onset for both high and low regularity sequences (Fig. 3c) (HR: 0–50 ms, $p < 0.0001$ cluster-corrected, $d = 0.78$; LR: first cluster 0–40 ms, $p = 0.007$ cluster-corrected, $d = 0.94$, second cluster 50–260 ms, $p < 0.0001$ cluster-corrected, $d = 1.72$). There was no significant difference in the prediction error encoding between the sequences ($p > 0.05$ cluster-corrected). Again, we repeated the analysis in a searchlight fashion across the cortical surface to investigate the cortical distribution. Prediction errors were encoded broadly across frontoparietal and temporal areas, with the right temporal cortex showing maximum neural prediction error signals in both sequences (Fig. 3c bottom).

## Error encoding correlates with representational shift

The above results unravel both, a neural signature of how the brain adapted its sensory representations to the predictability of inputs and how the brain encoded prediction errors. This allowed us to test our key hypothesis, i.e. that, during learning, stronger encoding of prediction errors is associated with stronger representational shifts. We focused our analysis on those two clusters of brain regions that showed a significantly stronger representational shift for the high as compared to low regularity sequence: left temporal cortex and

bilateral frontal cortices. Indeed, we found a significant positive correlation (Fig. 3d) between the magnitude of the prediction error and the representational shift in the left temporal cortex across subjects ($r = 0.47$, $p = 0.021$ Bonferroni-corrected). Frontal cortices showed no significant effect ($r = 0.22$, $p = 0.287$ Bonferroni-corrected). In sum, in accordance with our central hypothesis, we found that the stronger the prediction error signal was encoded in the left temporal cortex, the stronger the updating of the representational geometry in this brain region.

The prediction error trajectory from the ideal observer model followed an exponential decay (Fig. 3b, exponential/linear BIC = -4685.2/-4105.1). Thus, we investigated whether the representational dynamics during the high-regular sequence were also better explained by an exponential model as compared to a linear model. However, we found no statistically significant evidence supporting the exponential over the linear model, neither on the whole brain level (exponential/linear BIC = − 30.86/-29.99, $p = 0.165$, $d = 0.28$) nor in the left temporal (exponential/linear BIC = − 26.40/− 26.57, $p = 0.622$, $d = 0.09$) and frontal clusters (exponential/linear BIC = − 27.08/− 27.43, $p = 0.464$, $d = 0.14$).

### Synergistic large-scale encoding of prediction error

Finally, we aimed to extend our analysis framework beyond the traditional view of the brain as a collection of modular regions[45,46]. Thus, we asked if prediction error signals resulted from distributed processing across a network of brain regions rather than independent processing within each of these regions. We employed Partial Information Decomposition[37] to decompose the joint mutual information of pairs of brain regions about error signals 90–120 ms post tone onset into redundant and synergistic components[47,48] (Fig. 4a). This allowed us to investigate if networks of brain regions processed the prediction error either in a similar, but independent way (redundancy) or in a complementary, distributed fashion (synergy)[49].

We found that for both, high and low regularity conditions and across all pair-wise brain regions, the neural interactions encoding the prediction error were substantially synergistic, and this synergy was even significantly higher than the redundancy component (low regularity: $p = 0.035$, $d = 0.44$; high regularity: $p = 0.038$, $d = 0.43$). There was no significant difference in redundancy ($p = 0.549$) or synergy ($p = 0.502$) between low and high regularity conditions. To pinpoint which cortical interactions involved redundant and synergistic encoding, we contrasted these information components between the interval with the strongest prediction error encoding (90–120 ms post-tone onset) and the pre-tone baseline (− 50 to 0 ms) using network-based statistics (NBS)[50]. For both information components and regularity conditions, we found a large-scale network of interactions, involving mostly frontal, parietal, and temporal areas (Fig. 4c, views with connections) (all $p < 0.05$ component-corrected). Also, between-ness centrality, a measure of node importance, revealed frontoparietal and temporal cortices as hubs (Fig. 4c, views with shaded regions, all $p < 0.05$ cluster-corrected), with no significant difference between regularity conditions (interactions: all $p > 0.05$ component-corrected; betweenness centrality: all $p > 0.05$ cluster-corrected). In sum, we found that the encoding of prediction errors involved not only redundant but also synergistic interactions across a frontoparietal and temporal network.

### Synergistic error encoding correlates with representational shift

Do these synergistic interactions predict the representational shift during statistical learning? To address this, we repeated the same correlation analysis on the selected clusters that we performed on the 'independent' prediction error encoding and representational shift, but this time using the betweenness centrality of prediction error encoding synergy and redundancy (Fig. 4d, e). Indeed, we found that the centrality of the synergy of the left temporal cortex significantly correlated with the representational shift across participants (Fig. 4d, $r = 0.47$, $p = 0.022$ Bonferroni-corrected), while in bilateral frontal cortices, this was not the case ($r = 0.03$, $p = 0.902$ Bonferroni-corrected). For redundancy (Fig. 4e), neither the left temporal cortex ($r = 0.33$, $p = 0.113$ Bonferroni-corrected) nor bilateral frontal cortices ($r = 0.17$, $p = 0.434$ Bonferroni-corrected) showed a significant correlation. In sum, we found that the temporal cortex synergistically encoded prediction errors with a large-scale network of brain regions and that this synergistic encoding predicted the updating of sensory representations during learning.

## Discussion

Our results provide insights into how prediction error signals shape the representational geometry of the human brain. Specifically, our findings show a representational shift across time, through which sensory representations of contiguous and predictable tones became more similar. This effect is commonly referred to as chunking[51]. Our results accord well with a large number of studies that provided indirect evidence for chunking[22,38,52–57] as well as few previous studies that directly reported neural representations to be chunked according to the predictive structure of a sensory sequence[30,32]. Our result extends these findings based on a temporally resolved RSA[33], which allowed us to track the representational dynamics throughout learning. In line with previous work[20,30,32], our results revealed changes in the representational geometry consistent with chunking in both sensory and high-level brain regions. This suggests distinct computational systems across the cortical hierarchy tracking sensory statistics in parallel, possibly coordinated by hippocampal activity[32,58]. Notably, the temporally resolved RSA also allowed us to track the representational dynamics on a fast timescale for each tone presentation. This revealed an early latency of the chunking effect around 120 ms post-tone-onset, which temporally overlaps with the peak pitch representation. This suggests that once representational changes are established, they do not require further top-down modulation[59].

Similar to the neural processing sequences shown here, also the dynamics of deep neural networks display a chunking effect during training when gradient-based methods guide them in classification tasks[60,61]. Initially random in high-dimensional space, the hidden layers' representations become organised to distinguish class instances effectively through learning. This process mirrors the brain's processing strategy and suggests shared computational principles between natural and artificial systems[62–64]. Moreover, both systems might cluster similar sensory data to optimise energy use, adhering to environmental and computational limits, thus minimising extraneous exploration of the sensory state space in favour of more streamlined information processing[65–68].

Our results also shed light on the neural mechanisms underlying the cortical encoding of prediction errors. We exposed an ideal observer model[34,35,43,69] to the same sequences heard by human participants to extract theoretical precision-weighted prediction error trajectories. This computational approach allowed us to study prediction error encoding beyond the traditional comparison of standard and deviant stimuli in trial-based oddball paradigms[12,13,17,52] in a continuous, and thus more naturalistic, sequence paradigm[70]. Prediction error encoding peaked around 100 ms after each tone was delivered. As for the temporal dynamics of the representational shift, this relatively early latency in comparison to evidence reported on the mismatch negativity[12,17] could be ascribed to the specific experimental paradigm involving a continuous stimulus presentation. Also, the latency of the prediction error encoding temporally overlapped with pitch encoding. This accords well with recent evidence for the encoding of pitch and pitch expectations at similar latencies and different cortical sites in the human auditory cortex[71].

We found that error signals were encoded in a large-scale network involving temporal and frontoparietal cortices. Thus, in line with

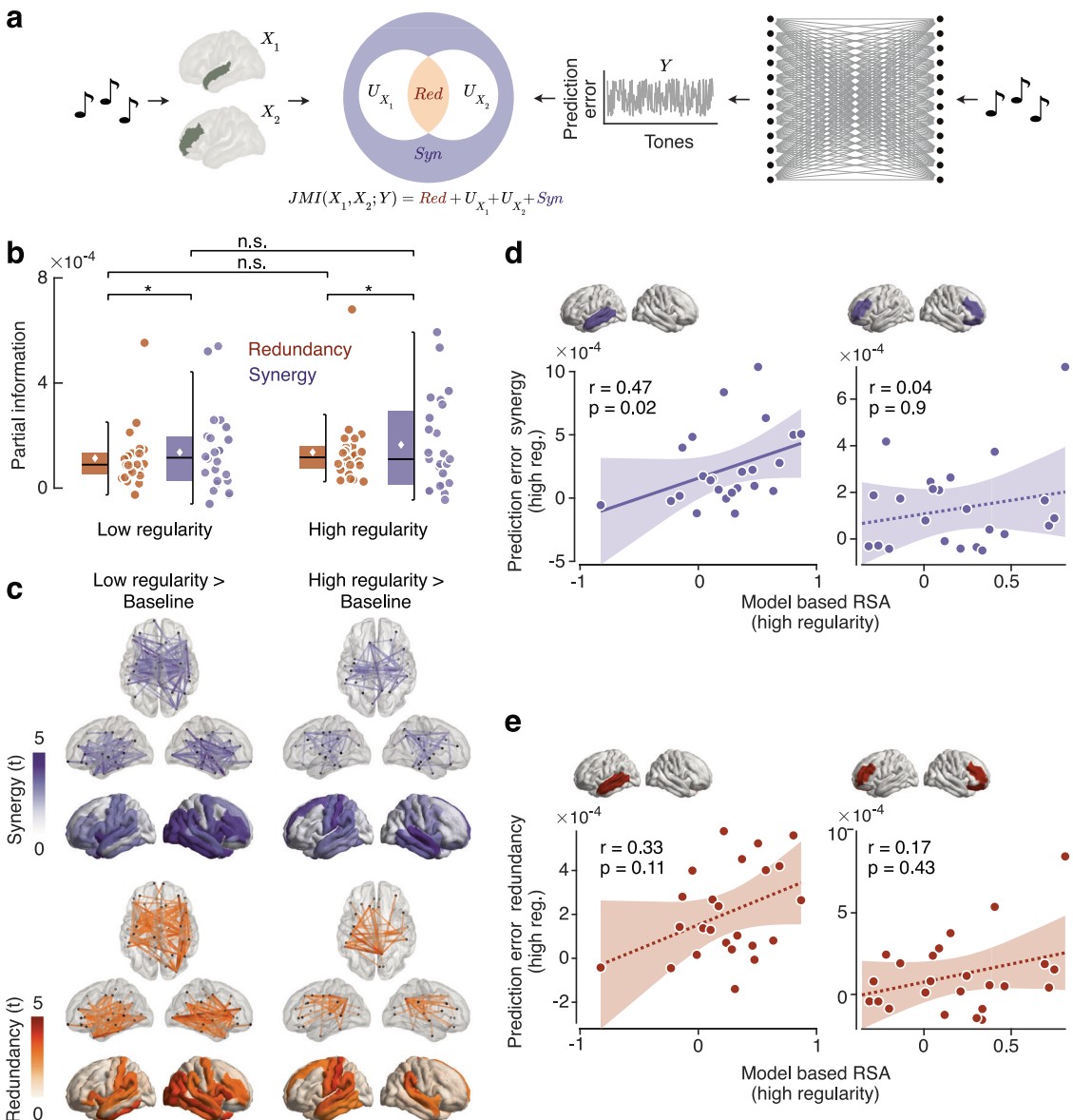

**Fig. 4 | I Partial Information Decomposition (PID) of the joint mutual information between pairs of brain areas encoding prediction errors. a** analysis framework for computing PID from the pairwise joint mutual information (centre), where the predictors ($X_1$ and $X_2$) are the brain signal across trials at the time window where the prediction error was mostly encoded (90–120 ms, left), and the predicted variable is the prediction error trajectory ($Y$, right). **b** average redundancy and synergy of prediction error encoding across all pairs of brain areas. Dots represent participants, and asterisks indicate statistical significance ($p < 0.05$). Significance was determined by paired two-tailed $t$ tests. **c** Redundancy and synergy t-statistics against the pre-tone baseline. Top views show significant cortical interactions. The bottom views show the betweenness centrality of each brain region. Transparency indicates statistical significance ($p < 0.05$ corrected), which was determined by component-based paired two-tailed permutation tests using network-based statistics (NBS). **d** correlation analysis between the representational shift and betweenness centrality of the synergy component for the brain regions indicated on the top. Dots represent individual participants. Shaded areas indicate 95% confidence intervals. The dotted lines indicate non-significant regression fits. Solid lines indicate significant regression fits. Significance ($p < 0.05$) was determined using the Pearson correlation coefficient and Bonferroni correction. Source data are provided as a Source Data file. **e** correlation analysis between the representational shift and betweenness centrality of the redundancy component. Dots represent individual participants. Shaded areas indicate 95% confidence intervals. The dotted lines indicate non-significant regression fits. Solid lines indicate significant regression fits. Significance ($p < 0.05$) was determined using the Pearson correlation coefficient and Bonferroni correction. Source data are provided as a Source Data file.

previous evidence, error signals were encoded at both sensory and high-level processing stages[35,72–74]. Importantly, we found no difference in prediction error encoding between low and high regularity conditions, suggesting that these brain areas encode the error signal in a context-independent manner, regardless of the volatility of environmental statistics[75,76].

The large-scale network of prediction error encoding led us to investigate how neural interactions between brain regions[77–79] contribute to error encoding. There is increasing evidence suggesting that cognitive and sensory processing in the brain is carried out in a distributed fashion rather than being localised[78,80–82] or, in other words, that the whole is greater than the sum of its parts[48]. To address this, we leveraged Partial Information Decomposition (PID)[37], which allows us to decompose the information dynamics between brain regions into synergistic and redundant interactions[49]. Here, redundancy implies that two brain regions encode the prediction error in the same way, indicating a common encoding mechanism[37,49,83]. In contrast, synergy reflects the tendency of the two brain regions to complementarily

encode the error signal, indicating a distributed encoding mechanism[37,48,49]. We found that, on average, the neural interactions encoding prediction errors were synergistic rather than redundant, suggesting a distributed rather than a shared encoding mechanism[37,48,49]. These findings provide evidence for synergistic interactions underlying the encoding of prediction errors in the human brain. These results add to a growing body of evidence that challenges the traditional view of the brain as a collection of modular areas[45,46] and suggests that predictive cortical processing is distributed rather than localised[48,78,80–82]. Furthermore, our results suggest that the dominance of synergistic encoding in the human brain is independent of contextual regularity. This is consistent with recent evidence from the marmoset brain suggesting that predictive processing is characterised by synergistic dynamics[47].

When we inspected neural interactions both at the network level and among the cortical hubs accounting for most of the information dynamics, we found that fronto-parietal and temporal regions were strongly interrelated by both redundant and synergistic interactions. Some of the strongest synergistic interactions involved the left and right auditory cortices, indicating that the two hemispheres can integrate information in a complementary fashion. Such pairwise interactions could be mediated by higher-order interactions involving other brain regions[84,85]. Importantly, the cortical distribution of synergistic encoding strongly overlapped with the results of modular searchlight analysis, comprising frontoparietal and temporal regions. Furthermore, the same areas that showed the correlation between error computation and representational shift were also cortical hubs of the synergistic interactions broadcasting the error signal throughout the cortex. This further supports the idea that prediction error encoding results from a network computation rather than local processing[45,86,87].

These findings posit possible challenges for the hypothesised hierarchical nature of predictive processing in some theories of predictive coding[6,21,23,74,88]. For example, in the Rao and Ballard model[6], the prediction error is computed independently at each hierarchical level. However, our results show that each brain region encodes only partial prediction error information with synergistic encoding across a large-scale network. Rather than isolated processes, this suggests feedback and recurrent processing of prediction error information within a network of brain regions[47]. Thus, our results support theories of predictive processing that do not necessarily require the hierarchical processing postulated by traditional predictive coding theories of perception[89,90].

Besides constraining the architectural aspects of predictive processing, our results provide critical evidence for the core hypothesis of the predictive coding framework, i.e. that the brain employs a generative model of the world and uses prediction errors to update model representations[4–6]. Our results provide direct evidence linking these two fundamental information processing primitives, prediction error encoding and representational change. We found that the brain areas that manifested a strong representational shift, at the same time, were synergistic hubs of prediction error encoding, corresponding to sensory areas such as the left auditory cortex. This may reflect a particular sensitivity of sensory areas to the modulation of their representational content[91,92]. At the same time, these areas could be an important target for top-down signals required for comparing sensory expectations and observations.

We found no conclusive evidence for non-linear representational dynamics that were aligned with the non-linear learning dynamics of our ideal observer model. On the one hand, this may merely reflect a lack of sensitivity. On the other hand, there could be fundamental differences between prediction error signalling and the resulting dynamics of representational updating. Alternatively, the chosen ideal observer model, which is the simplest model of predictive learning to accommodate our experimental paradigm, may not capture the individual learning dynamics. In any case, our results demonstrate the correlation between a linear trend of the representational shift and the strength of prediction error encoding. Future investigations may target the detailed shape of learning dynamics on the behavioural and neural levels.

In conclusion, our findings provide evidence that large-scale neural interactions engaged in predictive processing modulate the representational content of sensory areas, which may enhance the efficiency of perceptual processing in response to the statistical regularities of the environment.

## Methods

### Participants
Participants were 24 healthy volunteers (12 male) between 20 and 37 years old (mean age 27.54 years, SD = 9.96). All participants provided informed consent, were right-handed and had normal hearing abilities. The experiment was realised in accordance with the Helsinki Declaration and the local ethics committee of the Medical Faculty of the University of Tübingen approved the study (No. 231/2018BO1). Data were previously published in Moser et al.[38].

### Stimuli
Stimuli consisted of 12 pure sinusoidal tones between 261.63 and 932.33 Hz (Fig. 1a). The 12 tones coincided with the musical notes C, D, E, F#, G# and A# from the 4th and 5th octave of a standard piano (261.63, 293.66, 329.63, 369.99, 415.3, 466.16, 523.25, 587.33, 659.26, 739.99, 830.61, and 932.33 Hz). These tones served to create two sequences with different regularities. Both sequences consisted of 2400 tones (Fig. 1b, lasting 300 ms and presented every 333 ms) clustered into triplets (lasting 1 s) composed of three tones that never spanned more than one octave[38]. Each of the 12 tones appeared exactly 200 times in each of the two sequences. In each sequence, neither the same tone nor the same triplet of tones could repeat consecutively. In the high regularity sequence (Fig. 1c), there were only four types of triplets repeating over the course of the sequence. The order of tones inside a triplet was counterbalanced across participants, yielding three different combinations of the high regularity sequence. In the low regularity sequence (Fig. 1c), each triplet changed constantly throughout the sequence, albeit maintaining the octave constraint.

### Procedure
After completing a short hearing assessment with a screening audiometer (Hortmann Neuro-Otometrie Selector 20 K) to confirm normal hearing, participants were seated in a height-adjustable chair inside a magnetically shielded room and were told to fixate on a cross during the course of the experiment. Auditory stimulation was presented through earplugs at an intensity of 70 dB. Participants were instructed to passively listen to the sounds with no particular task to perform. The order of conditions was not counterbalanced, as the high regularity sequence always followed the low regularity sequence after a short break, for all participants[38].

### MEG data acquisition and pre-processing
MEG data were recorded using a 275-sensor, whole-head CTF MEG system (VSM Medtech, Port Coquitlam, Canada) installed in a magnetically shielded room (Vakuumschmelze, Hanau, Germany). The sampling rate MEG signal was 585.94 Hz. We first applied a fourth-order Butterworth band-pass filter (0.5–40 Hz) on the continuous data and then segmented the data from 0 to 330 ms relative to each stimulus onset correcting for a 32 ms sound onset delay relative to the recorded trigger signal[38]. Next, we resampled the data to 200 Hz and rejected noisy channels using a semi-automatic procedure, involving visual inspection and a cutoff threshold of root mean square (RMS) > 0.5 pT. We applied Independent Component Analysis (ICA)[93] to decompose the signal and discard eye movement, and muscular and cardiac artifacts, using FastICA[94] with the number of independent

components reduced to 50. The estimated independent components were visually inspected and rejected based on their topological, temporal and spectral characteristics whenever they showed an artifactual profile[95].

## Source reconstruction

After pre-processing, MEG sensors were aligned to the brain template "fsaverage"[96], from which we generated a single shell head model to compute the physical forward model[97] using FieldTrip[98]. Source coordinates, head model and MEG channels were co-registered on the basis of the nasion, left and right preauricular points. We used sensor-level MEG data, aggregated from both conditions, to estimate the filter weights of a Linearly Constrained Minimum Variance (LCMV) beamformer[40], with the regularisation parameter set to 5%. This spatial filtering approach reconstructs source activity with unit gain while, at the same time, maximising the suppression of contributions from other neural sources[40]. We fixed the orientation of the dipoles using singular value decomposition (SVD) to pick the direction that maximised the power[99]. Then, we used the filter weights to project single-trial MEG sensor data to the source space, correcting for the sign-flip due to the SVD applied for selecting the optimal orientation that maximises output power. The source space was finally parcelled into 72 different brain areas using the Desikan-Killany parcellation scheme[39].

## Representational Similarity Analysis

We first analysed source-reconstructed MEG using a Representational Similarity Analysis (RSA) approach. We split the trials (each tone) into 5 non-overlapping blocks over the course of each sequence and computed the representational dissimilarity matrices (RDMs) at each block separately (Fig. 2a). We used the cross-validated Mahalanobis distance (cvMD)[41,42] as a dissimilarity metric with a 10-fold cross-validation scheme, due to its renowned statistical properties especially suited for RSA with neuroimaging data[41,100]. We applied the Ledoit-Wolf method[101] to compute the asymptotically optimal shrinkage parameter to regularise the covariance matrix from the training set. We ordered the RDMs entries in both sequences to have the diagonal representing the distance between tones belonging to each triplet in the high regularity sequence (Fig. 2a). The rows of the RDMs corresponded to the first, second and third tone in each triplet, ordered from the first to the fourth triplet (e.g., the fourth row was the first tone of the second triplet, which could have been either tone $D_4$, $G\#_4$ or $C_5$). RDMs were computed either in a time-resolved fashion[102], using all brain parcels as features for each time point, or in a searchlight manner[103], using as a feature each brain parcel alongside its 5 spatial nearest neighbours for a certain time window (averaging across the time dimension). Then, we averaged the values on the diagonal to investigate how, through learning, the tones within a triplet distanced between each other, as well as off the diagonal, to study the same effect between tones that did not belong to the same triplet. To summarise this pattern, we adopted a model-based RSA[104] by designing an RDM which had zeros on the on-diagonal entries and ones on the off-diagonals. We fitted this model-based RDM to each brain-derived RDM using Spearman correlation. Finally, we computed the slope of the on-diagonal, off-diagonal and model-based estimates across the blocks using the ordinary least square estimator with a counting vector increasing from 1 to 5 as a regressor.

## Ideal observer model

We fitted prediction error trajectories extracted from an ideal observer model to the source-reconstructed MEG data. The employed ideal observer model can be conceived as a perceptron neural network (Fig. 3a), receiving as input one tone at a time and attempting to predict the next one. We represented the stimuli categorically as a one-hot encoded vector of the same length as the number of different tones

used in the acoustic sequences $x_t \in \mathbb{R}^{1 \times n}$, where $n$ is equal to 12 and $t$ indexes the tones in a sequence. Therefore, the input layer had the same dimensionality as the output layer. The trainable model parameters were encoded in a weight matrix $W_t \in \mathbb{R}^{n \times n}$, which connected the input and output layers. The weight matrix was initialised as a uniform prior over the categorical distribution of the tones, with all values having $n^{-1}$ as entries. Given each tone, the model predicted the next one according to the following equation:

$$z_t = x_t \cdot W_t \tag{1}$$

$$\hat{y}_t^i = e^{z_t^i} \left( \sum_{i=1}^n e^{z_t^i} \right)^{-1} \tag{2}$$

where $\hat{y}_t$ represents the prediction of the model for the next tone. We defined the loss function $\mathcal{L}$ as the maximum likelihood or cross-entropy function:

$$\mathcal{L}(x_{t+1}, \hat{y}_t) = \sum_{i=1}^n x_{t+1}^i \log(\hat{y}_t^i) \tag{3}$$

The model was trained by passing all the tones from one sequence at a time, and after each observation, we computed the partial derivative of the loss function with respect to $W_t$:

$$\frac{\partial \mathcal{L}}{\partial W_t} = x_t^T \cdot (\hat{y}_t - x_{t+1}) \tag{4}$$

This gradient, combined with a dynamic learning rate parameter, gave rise to our measure of prediction error:

$$\omega = \sum_{i=1}^n \hat{y}_t^i \log(\hat{y}_t^i) \tag{5}$$

$$PE_t = \omega \frac{\partial \mathcal{L}}{\partial W_t} \tag{6}$$

Here, the learning rate $\omega$ is not fixed as in classical reinforcement learning models[34] but depends on the uncertainty of the model since it is the Shannon entropy of the predictive distribution. This precision-weighted prediction error can account for learning phenomena better than classical models with fixed learning rate[43,105]. Finally, we updated the parameters $W_t$ using the gradient descent algorithm:

$$W_{t+1} = W_t - PE_t \tag{7}$$

This model can also be viewed as a categorical and dynamic version of the Rescorla-Wagner model[34]. We extracted the trajectory of the prediction errors separately for each sequence and fitted it to each parcel of the brain, for each participant. To fit the prediction error trajectories, we adopted the Gaussian Copula Mutual Information (GCMI) method (Fig. 3a), a robust multivariate statistical framework that combines the statistical theory of copulas with the analytical solution for the Shannon entropy computation of Gaussian variables[44]. We first transformed each variable (brain data and prediction error trajectories) into a Gaussian variable using the inverse normal transformation. For each variable under consideration, the transformed value was obtained as the inverse standard normal cumulative distribution function (CDF) evaluated at the empirical CDF value of that variable[44]. After this procedure, the mutual information is computed

parametrically for Gaussian variables as follows:

$$\mathrm{MI}(X;Y) = \frac{1}{2\ln 2}\ln\left[\frac{|\Sigma_X||\Sigma_Y|}{|\Sigma_{XY}|}\right] \quad (8)$$

where $\Sigma_X$ and $\Sigma_Y$ are the covariance matrices of $X$ and $Y$, respectively and $\Sigma_{XY}$ is the covariance matrix for the joint variable $(X, Y)$. In our study, we considered $Y$ always as the (univariate) prediction error trajectory and $X$ as the multivariate brain data. The inverse normal transformation for the brain data was applied to each feature univariately. GCMI values were computed either in a time-resolved fashion[102], using all brain parcels as variables for each time point, or in a searchlight manner[103], using as $X$ each brain parcel alongside its 5 spatial nearest neighbours for a certain time window (averaging across the time dimension). Finally, we baseline-corrected these values by subtracting the GCMI values computed before the onset of the stimulus $x_t$ in the time window from $-50$ ms to $0$ ms, to avoid possible confounds given by the autocorrelation function of the two sequences.

## Partial Information Decomposition

We analysed the MEG data using an information-theoretic approach, to investigate how brain areas interact when jointly encoding the prediction error signal. We employed Partial Information Decomposition (PID)[37] to decompose the joint mutual information (JMI), which is the information that two variables $X_1$ and $X_2$ give about a third target variable $Y$, in terms of different kinds of informational atoms:

$$\mathrm{JMI}(X_1, X_2; Y) = R + U_{X_1} + U_{X_2} + S \quad (9)$$

information provided by one variable but not the other (denoted as unique information $U_{X_1}$ or $U_{X_2}$), information provided by both variables separately (redundant information $R$), or jointly by their combination (synergistic information, $S$). In this study, we considered $Y$ always as the prediction error trajectory and $X_1$ and $X_2$ as pairs of brain parcels for each time point in a certain time window. Thus, following the intuition of the PID framework[37], we computed the redundancy measure as the minimum intersection in the information ($I_{\min}$) provided by both $X_1$ and $X_2$ about $Y$ as follows:

$$R = I_{\min}(X_1, X_2; Y) = \sum_{y\in Y} \min_{x_i}\left(\sum_{i=1}^{2} p(x_i, y)\log\left(\frac{p(x_i, y)}{p(x_i)p(y)}\right)\right) \quad (10)$$

Finally, all the remaining terms can be computed using linear algebra as follows:

$$U_1 = \mathrm{MI}(X_1; Y) - I_{\min}(X_1, X_2; Y) \quad (11)$$

$$U_2 = \mathrm{MI}(X_2; Y) - I_{\min}(X_1, X_2; Y) \quad (12)$$

$$S = \mathrm{JMI}(X_1, X_2; Y) - U_1 - U_2 - I_{\min}(X_1, X_2; Y) \quad (13)$$

All these quantities were computed by first normalising the variables using the same procedure as above for the GCMI (i.e., the inverse normal transformation). This allowed us to compute a closed form of the quantities following a parametric Gaussian model[44,106]. We computed PID components for all combinations of two brain parcels and extracted only the redundancy and synergy terms, separately for each sequence. This procedure yielded two adjacency matrices of $72 \times 72$ representing the pairwise neural interactions encoding the prediction error, one for the redundant and the other for the synergistic interactions. Again, we baseline-corrected these values by subtracting the redundancy and synergy computed in the baseline. Then, we averaged the matrices as a measure of the global efficiency of the redundant and synergistic network, thus yielding one value per participant and

sequence. Finally, we also computed the betweenness centrality measure as a measure of node importance by marginalising the redundancy and synergy values across the interaction dimension.

## Statistical analysis

All statistical analyses were carried out at the group level (random effects) using mass univariate cluster-based paired two-tailed permutation tests based on t-statistics[107] with a significance threshold α set to 0.05, 10 000 iterations, maxsum as cluster statistic and the topological neighbourhood structure defined by the proximity of the brain parcels. For the grand average redundancy and synergy, we carried out paired two-tailed $t$ tests, while for the statistical comparison of adjacency matrices, we used network-based statistics (NBS)[50] with a significance threshold α set to 0.05, 10 000 iterations, and the size of the connected component as component statistic. For the correlation analyses, we computed the right-tail Pearson's correlation coefficient for each selected brain cluster between the representational shift effect and the encoding prediction error effect (GCMI, redundancy and synergy centrality) across participants and correcting the multiplicity by using the Bonferroni correction. The brain clusters were selected using the first step of the cluster-based permutation test used above, i.e. by selecting the contiguous brain parcels that surpassed the alpha threshold set above (0.05). For the model fitting of the representational dynamics, we used an exponential model of the form $y = ae^{-bx} + c$, with $a$, $b$ and $c$ as free parameters. The linear model had the intercept and slope as free parameters. Both models were fitted using the non-linear least square method with no bounds on the parameter space and resampling 100 times the starting points. We computed the Bayesian Information Criterion (BIC) for model comparison at the subject level and tested the winning model with a paired two-tailed $t$ test.

## Reporting summary

Further information on research design is available in the Nature Portfolio Reporting Summary linked to this article.

# Data availability

Source data are provided in this paper. Raw MEG data to reproduce all the results in our study are openly available at Zenodo via the following link: https://doi.org/10.5281/zenodo.3961467. Source data are provided with this paper.

# Code availability

Data analysis was carried out in MATLAB version 2022a. We used the open-source library Fieldtrip (version 20220827) available at https://github.com/fieldtrip/fieldtrip for MEG analysis. We also used the GCMI library (version 1.0) for information-theoretic analyses openly available at https://github.com/robince/gcmi.

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

## Acknowledgements

This study was supported by the European Research Council (ERC; https://erc.europa.eu/) CoG 864491 (M.S.), by the German Research Foundation (DFG; https://www.dfg.de/) projects 276693517 (SFB 1233) (M.S.) and SI 1332/6-1 (SPP 2041) (M.S.), and by the German Federal Ministry of Education and Research (BMBF) to the German Centre for Diabetes Research (DZD01GI0925, H.P.).

## Author contributions

A.G.: Conceptualisation, Software, Formal analysis, Visualisation, Writing – original draft, Writing – Review & Editing. J.M.: Methodology, Investigation, Data curation, Software, Writing – Review & Editing. H.P.: Methodology, Supervision, Writing – Review & Editing. M.S.: Conceptualisation, Supervision, Resources, Project administration, Funding acquisition, Writing – original draft, Writing – Review & Editing.

## Funding

## Competing interests

The authors declare no competing interests.
