## [Transparent Peer Review file · Nature Communications]

Predictive learning shapes the representational geometry of the human brain

Corresponding Author: Dr Antonino Greco

Version 0:

Reviewer comments:

Reviewer #1

(Remarks to the Author)

This is an interesting re-analysis of data published earlier. The results bring new knowledge on the prediction error encoding in the human brain by showing changes occurring during the experiment and on how the prediction error information is spread in a synergistic manner across brain regions.

Although the manuscript is written clearly and soundly in general, I have some suggestions for improvement below:

Line 66. "Furthermore, studies on statistical learning show that the similarity of neuronal representations of sensory stimuli reflects the learned statistical dependencies between these stimuli" -> Please specify what is meant by neuronal representations here.

Line 68. "This suggests that statistical learning may shape the geometry of neural representations to match the geometry of sensory inputs. However, the neural mechanisms underlying this learning remain unclear." Please open what is meant by geometry here. The next sentence is defining this to some degree, but it is not clear if there is one-to-one correspondence here.

Line 85: listed -> listened

Line 97: Killany -> Killiany

Line 234-236: "It is important to note that this effect may reflect the naturalistic and continuous presentation of stimuli in our paradigm. Different outcomes may arise in segmented, trial-based settings." Could this be elaborated, why there could be a difference between continuous versus trial-based settings and what kind of differences would be expected?

In several sections of the manuscript the paradigm is described as naturalistic. However, it is still using repeated tones that are irrelevant to the participants with very limited link to real-life situations. Please either rephrase or justify the description of the paradigm.

Line 217-218: The first sentence on its own does not hold up as its own paragraph.

Line 247-249: "We exposed an ideal observer model (30, 31, 39, 65) to the same sequences seen by human participants to extract theoretical precision-weighted prediction error trajectories." -> Should this be heard instead of seen?

Line 291-292: "Importantly, the cortical distribution of synergistic encoding strongly overlapped with the results of modular searchlight analysis." Please describe in more detail how exactly were they overlapping.

Line 300-303: "However, the large-scale synergistic component in our results suggests that each brain region encodes only partial information needed for the error computation, which is then broadcasted into the network and integrated with feedback and recurrent connections to eventually process the prediction error (43)." Please elaborate which part your results support this and which part is speculation based on earlier studies.

Line 327. Would it be possible to mention already earlier in the introduction that the data were published, analysed and interpreted in a previous paper? Also, it would be good to see discussion on the findings of the previous analyses and the current re-analysis.

Line 348. Could the lack of counterbalancing of the conditions have affected the results?

Line 355. "...and then selecting all the tones from 0 to 330 ms" It is unclear what was done here with the filter?

Reviewer #2

(Remarks to the Author)

This study investigates how the brain's predictive learning impacts its representation of sensory information. Using magnetoencephalography (MEG) to monitor participants exposed to different auditory sequences, researchers discovered that the brain adjusts its representation structures to match the statistical properties of the sensory inputs. This alignment results in the clustering of representations of predictable stimuli. They noted that the degree of this alignment in sensory areas correlated with the strength of prediction error encoding.

Overall, I found the study to provide a thorough analysis of an interesting and important dataset, carried out to a high standard. I consider the conclusions to be supported by the results and analysis. This work is interesting and innovative. However, while reading, several major and minor questions arose that the authors might want to answer.

Major

Definition of learning: The most important aspect to question is the learning effect. The title and general form of the article imply that participants learned something. However, due to passive stimulation, no conclusions about learning effects in terms of behavioral consequences can be drawn. The article would benefit if the authors clearly define what predictive learning is, such as the reduction of prediction error representation through repeated experience with statistical regularities in the environment.

If this is what the authors mean, then the progression of the prediction error (Fig. 3b) as the learning model should be presented earlier. It clearly shows the dynamics with which the prediction error is reduced in an ideal observer. This is also a good representation of the learning dynamics. However, the authors might want to reconsider their approach then. This is because the learning/prediction error reduction is not linear. The learning dynamics shown in Figure 3b indicate that learning mainly takes place within the first tenth and then no longer. Therefore, the model assumption in the linear correlation analysis (Fig. 2) may not be accurate. On the bright side, the progression in Fig. 2c over the blocks in the high regularity condition does indeed not look linear (a strong change between block 1 and 2 followed by little change over the remaining blocks). If this is the case, it would actually strengthen the authors' main point. The authors could test this by showing that the data follows a similar exponential decrease that corresponds to the change in prediction error.

Activation patterns I do not understand how the source activation comes about. Were the responses initially selected at the sensor level and then projected to the source level? If yes, what do the topography and the actual response look like? If the largest activation is in the sTG, as Figure 1b suggests, why were the responses taken bilaterally from the entire temporal cortex, as the small insets suggest? Also, the authors show the cortical distribution of responses between 50 and 70 msec post-onset (Figure 1b). However, the peak decoding performance is around 100 msec, and all later reported effects are around 120 msec. I think the spatial distribution around 120 msec would also be important. Furthermore, the prediction error encoding around 100-120 msec might fall in the temporal interval of a. Hence, do the authors basically evaluate a mismatch response? Can the authors simply show a response to all tones in the high and low regularity conditions to see if this is a mismatch response?

Minor

- Figure 1 shows a decrease along with a decrease in standard error. How does that happen? Were these time courses concatenated? It doesn't seem to be ongoing activity.
- It is not explicitly stated, or at least I did not find it. Are the 12 tones similarly distributed in both conditions? Please provide a graphic showing this.
- I don't understand the transition matrices in Figure 1a. The simplest explanation is that it is upside down. I assume each row stands for one of the twelve tones. The color codes indicate how likely one of the next tones is. Thus in the high regularity condition (lower panel), according to the transition matrix, tone 4 always follows tone 1. However, this is not correct given the triplet representation on the left. The highest tone (932Hz) is always followed by the second highest tone in the tone frequency representation. Therefore, the yellow field (100%) should be at matrix position (1,2). The figure only makes sense if the bottom row refers to the highest tone and the top row to the lowest tone. I also assume that the transition matrices show the statistics across the entire experiment given that in the upper panel transition probabilities in the triplet representation do not match transition probabilities in the matrix. For example, I count 9 tones with a frequency of 262 Hz. Four out of these nine tones are followed by tone 2. This results in a probability of 44%. But the color code suggests that 25% is the maximal transition probability.

At some points, I found the writing unnecessarily complicated. For example, "the betweenness centrality of synergy and redundancy revealed the frontoparietal and temporal cortices as hubs for both information components."

Then the authors write (line 234) that this paradigm represents a naturalistic presentation. The paradigm is indeed less artificial than an oddball paradigm, but it is still far from a naturalistic setting.

By addressing these points, the manuscript will become clearer and more accessible to readers, enhancing the impact and understanding of this important work.

Version 1:

Reviewer comments:

Reviewer #1

(Remarks to the Author)

The authors have responded to all of my comments and addressed all the issues raised.

Reviewer #2

(Remarks to the Author)

The authors have addressed my concerns in a clear and detailed manner, making several changes to the manuscript. First, they provided a definition of predictive learning in the introduction and presented behavioral results from a prior study using the same dataset, addressing my concern about the lack of behavioral data. This demonstrated evidence of learning through passive exposure. Additionally, they moved the presentation of the prediction error model earlier in the manuscript, as requested, and explained that learning dynamics were better modeled with an exponential rather than a linear approach. Although they found no significant improvement with the exponential model, they reported these findings transparently.

Regarding activation patterns, I had requested more clarity on the source-level analysis and the time windows used. The authors responded by clarifying their methods and adding additional spatial distribution plots for the 100-120 ms window, as requested. They also addressed the potential for mismatch responses but found no significant differences between high- and low-regularity conditions. For minor comments, such as issues with Figure 1 and the transition matrices, the authors made the necessary corrections and clarified that the tones were equally distributed in both conditions. Overall, the authors' revisions have made the manuscript more comprehensive and effectively addressed my concerns.

We thank the reviewers and the editor for their positive assessment of our work, and for their thoughtful comments. Following the reviewers' advice, we performed new analyses, revised the entire manuscript and added further detail and clarification. We are confident that these changes address all concerns of the reviewers and substantially strengthen the manuscript.

Main changes and new results:

- We performed and added a new analysis investigating the alignment between the dynamics of the ideal observer model and the representational dynamics in the brain.
- We report behavioral results from a previous study on this dataset that show statistical learning at the behavioral level.
- We added a new supplementary figure showing the evoked activity in an additional time window.
- We thoroughly revised the text and added explicit definitions of major theoretical construct such as “predictive learning” or “neural representation”.
- We now comply with all editorial and formatting guidelines as suggested by the editor.

Below, we provide point-by-point responses to all reviewers. To ease navigation, reviewer comments are in **black** and our replies in **blue**.

Reviewer 1

This is an interesting re-analysis of data published earlier. The results bring new knowledge on the prediction error encoding in the human brain by showing changes occurring during the experiment and on how the prediction error information is spread in a synergistic manner across brain regions.

Although the manuscript is written clearly and soundly in general, I have some suggestions for improvement below:

We thank the reviewer for the positive assessment of our work and for his/her thoughtful comments, which were very helpful for improving the manuscript.

Line 66. "Furthermore, studies on statistical learning show that the similarity of neuronal representations of sensory stimuli reflects the learned statistical dependencies between these stimuli" -> Please specify what is meant by neuronal representations here.

We followed the reviewers suggestion and now specify what we referred to as neuronal representations. The revised section reads: "[...] Furthermore, studies on statistical learning show that the similarity of neuronal representations of sensory stimuli, i.e. of neural activity patterns that are specific for distinct stimuli (26), reflects the learned statistical dependencies between these stimuli (27–29). [...]" (line 59)

Line 68. "This suggests that statistical learning may shape the geometry of neural representations to match the geometry of sensory inputs. However, the neural mechanisms underlying this learning remain unclear." Please open what is meant by geometry here. The next sentence is defining this to some degree, but it is not clear if there is one-to-one correspondence here.

We followed the reviewers suggestion. The revised section now reads: “[...] This suggests that statistical learning may shape the geometry, i.e. mutual similarities, of neural representations to match the geometry, i.e. mutual statistical dependencies, of sensory inputs. [...]” (line 61)

Line 85: listed -> listened Line 97: Killany -> Killiany

We thank the reviewer for spotting these typos. We corrected them.

Line 234-236: "It is important to note that this effect may reflect the naturalistic and continuous presentation of stimuli in our paradigm. Different outcomes may arise in segmented, trial-based settings." Could this be elaborated, why there could be a difference between continuous versus trial-based settings and what kind of differences would be expected?

We thank the reviewer for pointing out this lack of clarity. We decided to omit this point and to avoid speculation.

In several sections of the manuscript the paradigm is described as naturalistic. However, it is still using repeated tones that are irrelevant to the participants with very limited link to real-life situations. Please either rephrase or justify the description of the paradigm.

We thank the reviewer for bringing up this point. We agree with the reviewer that our stimuli are still far from being natural or naturalistic. We used the term ‘naturalistic’ merely to highlight that the present continuous stimulation paradigm is more naturalistic than traditional trial-based odd-ball paradigms. To avoid any potential misunderstanding, we removed all uses of the term except for one case where we explicitly use it in a comparative sense: “[...] This computational approach allowed us to study prediction error encoding beyond the traditional comparison of standard and deviant stimuli in trial-based oddball paradigms (10, 11, 15, 49) in a continuous, and thus more naturalistic, sequence paradigm (67). [...]” (line 257).

Line 217-218: The first sentence on its own does not hold up as its own paragraph.

We followed the reviewers suggestion and attached the sentence to the following paragraph.

Line 247-249: "We exposed an ideal observer model (30, 31, 39, 65) to the same sequences seen by human participants to extract theoretical precision-weighted prediction error trajectories." -> Should this be heard instead of seen?

We followed the reviewers suggestion and changed the verb.

Line 291-292: "Importantly, the cortical distribution of synergistic encoding strongly overlapped with the results of modular searchlight analysis." Please describe in more detail how exactly were they overlapping.

We followed the reviewers suggestion. The revised section reads: “[...] Importantly, the cortical distribution of synergistic encoding strongly overlapped with the results of the modular searchlight analysis, comprising frontoparietal and temporal regions. [...]” (line 299)

Line 300-303: "However, the large-scale synergistic component in our results suggests that each brain region encodes only partial information needed for the error computation, which is then broadcasted into the network and integrated with feedback and recurrent connections to eventually process the prediction error (43)." Please elaborate which part your results support this and which part is speculation based on earlier studies.

We thank the reviewer for pointing out this lack of clarity. We followed the reviewers suggestion and rephrased this section to specify what are our results and what are speculations. The revised section reads: “[...] However, our results show that each brain region

encodes only partial prediction error information with synergistic encoding across a large-scale network. Rather than isolated processes, this suggests feedback and recurrent processing of prediction error information within a network brain regions (44). [...]” (line 307)

Line 327. Would it be possible to mention already earlier in the introduction that the data were published, analysed and interpreted in a previous paper? Also, it would be good to see discussion on the findings of the previous analyses and the current re-analysis.

We followed the reviewers suggestion and added a section in the results where we elaborate on the previous behavioral findings related to this dataset, which reads: “[...] A previous behavioral analysis on the dataset showed that participants learned the statistical regularities of the tone sequences³⁵. After listening to both stimulus sequences, participants rated triplets from the high regularity sequence as significantly more familiar than partial or random triplets. Thus, participants showed significant statistical learning through passive stimulus exposure.” (line 93)

Concerning the question at which point to mention the previous publication, we opted not to do this already in the introduction, because we felt this would be distractive as the main focus and question of the present paper is very different from the previous publication. The previous use of this data is now explicitly mentioned at the beginning of the methods section and in the results section.

Line 348. Could the lack of counterbalancing of the conditions have affected the results?

We thank the reviewer for bringing up this point. We are confident that this is not the case, especially regarding our main finding of a link between prediction error encoding and the representational shift effect. The reason is that all our main hypotheses are tested within the high regularity sequence, without comparing it to the low regularity sequence. Therefore, we are confident that not counterbalancing the order of the sequences has no effect on these results.

Line 355. "...and then selecting all the tones from 0 to 330 ms" It is unclear what was done here with the filter?

We thank the reviewer for bringing up this point. We epoched the data after the filtering operation that was applied on the continuous recordings. We revised the text to clarify this: “[...] We first applied a fourth-order butterworth band-pass filter (0.5 to 40 Hz) on the continuous data and then segmented the data from 0 to 330 ms relative to each stimulus onset correcting for a 32 ms sound onset delay relative to the recorded trigger signal (35). [...]” (line 371)

Reviewer 2

This study investigates how the brain's predictive learning impacts its representation of sensory information. Using magnetoencephalography (MEG) to monitor participants exposed to different auditory sequences, researchers discovered that the brain adjusts its representation structures to match the statistical properties of the sensory inputs. This alignment results in the clustering of representations of predictable stimuli. They noted that the degree of this alignment in sensory areas correlated with the strength of prediction error encoding.

Overall, I found the study to provide a thorough analysis of an interesting and important dataset, carried out to a high standard. I consider the conclusions to be supported by the results and analysis. This work is interesting and innovative. However, while reading, several major and minor questions arose that the authors might want to answer.

We thank the reviewer for the positive assessment of our work and for the thoughtful comments, which were very helpful for improving the manuscript.

Major

Definition of learning: The most important aspect to question is the learning effect. The title and general form of the article imply that participants learned something. However, due to passive stimulation, no conclusions about learning effects in terms of behavioral consequences can be drawn. The article would benefit if the authors clearly define what predictive learning is, such as the reduction of prediction error representation through repeated experience with statistical regularities in the environment.

We thank the reviewer for bringing up this very important point. Indeed, a previous study on these data showed significant behavioral learning effects. We now report these important results early in the results section of the revised manuscript: “[...] A previous behavioral analysis on the dataset showed that participants learned the statistical regularities of the tone sequences (35). After listening to both stimulus sequences, participants rated triplets from the high regularity sequence as significantly more familiar than partial or random triplets. Thus, participants showed significant statistical learning through passive stimulus exposure.” (line 93)

Moreover, we followed the reviewer’s suggestion and added an explicit definition of predictive learning in the introduction, which now reads: “[...] We defined predictive learning as the process of minimizing prediction errors about future sensory observations and employed computational modelling to derive neural signals encoding prediction error trajectories (31–33). [...]” (line 73)

If this is what the authors mean, then the progression of the prediction error (Fig. 3b) as the learning model should be presented earlier. It clearly shows the dynamics with which the prediction error is reduced in an ideal observer. This is also a good representation of the learning dynamics. However, the authors might want to reconsider their approach then. This is because the learning/prediction error reduction is not linear. The learning dynamics shown in Figure 3b indicate that learning mainly takes place within the first tenth and then no longer. Therefore, the model assumption in the linear correlation analysis (Fig. 2) may not be accurate. On the bright side, the progression in Fig. 2c over the blocks in the high regularity condition does indeed not look linear (a strong change between block 1 and 2 followed by little change over the remaining blocks). If this is the case, it would actually strengthen the authors' main point. The authors could test this by showing that the data follows a similar exponential decrease that corresponds to the change in prediction error.

We thank the reviewer for bringing up this important and interesting point. We agree that the prediction error dynamics of our ideal observer model is non-linear. Indeed, a model comparison analysis revealed a better description of the prediction error dynamics by an exponential model as compared to a linear model. Thus, we followed the reviewer’s suggestion and investigated if also the representational dynamics were significantly better described by an exponential than by a linear model. However, this was not the case. Neither for a whole brain analysis, nor for the individual cortical clusters of interest, an exponential

model was significantly better than a linear model (all $p > 0.05$). Despite these negative findings and inspired by the reviewer's comment, we felt that these results are informative and decided to include them in the revised manuscript along with their discussion. The corresponding results section now reads: "The prediction error trajectory from the ideal observer model followed an exponential decay (Fig. 3B, exponential/linear BIC = -4685.2/-4105.1). Thus, we investigated whether the representational dynamics during the high-regular sequence were also better explained by an exponential model as compared to a linear model. However, we found no statistically significant evidence supporting the exponential over the linear model, neither on the whole brain level (exponential/linear BIC = -30.86/-29.99, $p = 0.165$, $d = 0.28$), nor in the left temporal (exponential/linear BIC = -26.40/-26.57, $p = 0.622$, $d = 0.09$) and frontal clusters (exponential/linear BIC = -27.08/-27.43, $p = 0.464$, $d = 0.14$)." (line 182).

The corresponding section in the discussion reads: "We found no conclusive evidence for non-linear representational dynamics that were aligned with the non-linear learning dynamics of our ideal observer model. On the one hand, this may merely reflect a lack of sensitivity. On the other hand, there could be fundamental differences between prediction error signalling and the resulting dynamics of representational updating. Alternatively, the chosen ideal observer model, which is the simplest model of predictive learning to accommodate our experimental paradigm, may not capture the individual learning dynamics. In any case, our results demonstrate the correlation between a linear trend of the representational shift and the strength of prediction error encoding. Future investigations may target the detailed shape of learning dynamics on the behavioral and neural level." (line 326)

Activation patterns I do not understand how the source activation comes about. Were the responses initially selected at the sensor level and then projected to the source level? If yes, what do the topography and the actual response look like? If the largest activation is in the sTG, as Figure 1b suggests, why were the responses taken bilaterally from the entire temporal cortex, as the small insets suggest? Also, the authors show the cortical distribution of responses between 50 and 70 msec post-onset (Figure 1b). However, the peak decoding performance is around 100 msec, and all later reported effects are around 120 msec. I think the spatial distribution around 120 msec would also be important. Furthermore, the prediction error encoding around 100-120 msec might fall in the temporal interval of a. Hence, do the authors basically evaluate a mismatch response? Can the authors simply show a response to all tones in the high and low regularity conditions to see if this is a mismatch response?

We thank the reviewer for bringing up this point. Figure 1b was only meant to provide a sanity check that our recordings and source-reconstruction approach yields robust sensory signals. Thus, we simply used the bilateral temporal cortices as defined in the Desikan-Killiany parcellation scheme to test if temporal areas showed robust neural responses, as one would expect for auditory stimulation. We clarified this in the revised manuscript. Moreover, we agree with the reviewer that the spatial distribution around 100-120 ms is also interesting to show. Thus, we added this plot in a new supplementary figure. The revised manuscript now reads: "[...] As expected, both tone sequences evoked neural responses that peaked about 60 ms after each tone onset in bilateral auditory cortices (Fig. 1b) and at about 110 ms in left temporal cortex (Fig. S1). Thus, source-reconstruction yielded robust cortical responses. We next applied [...]" (line 100).

Concerning the comparison between low- and high- regular conditions we are not entirely sure about the reviewer's question because there is an incomplete sentence ("[...] might fall in the

temporal interval of a.”). We assume that the reviewer is asking if the investigated signals correspond to a mismatch response. Indeed, the mismatch response has been interpreted as a neural signature of prediction error encoding, but it is usually investigated with trial-based oddball paradigms. Here, we used a more nuanced paradigm in which prediction errors were computed in a continuous auditory stream. In any case, we followed the reviewer’s suggestion and directly compared responses between the high regularity and low regularity conditions 100-120 ms after tone onset averaging across all tones. As for all other analysis we performed this analysis in source space using a cluster-based permutation statistic. We did not find any significant difference between conditions across the cortex. Thus, this analysis yielded no evidence that the 100-120 ms response is a mismatch response.

Minor

Figure 1 shows a decrease along with a decrease in standard error. How does that happen? Were these time courses concatenated? It doesn't seem to be ongoing activity.

We thank the reviewer for spotting this. These data points were indeed technical artifacts from concatenating the data segments. This happened because we first epoched the data and then resampled them to 200 Hz. We now corrected this in the revised manuscript.

It is not explicitly stated, or at least I did not find it. Are the 12 tones similarly distributed in both conditions? Please provide a graphic showing this.

We thank the reviewer for bringing up this point. Indeed, all tones had the same overall occurrence (200 times) in both sequences. We added this information in the revised manuscript (lines 354).

I don’t understand the transition matrices in Figure 1a. The simplest explanation is that it is upside down. I assume each row stands for one of the twelve tones. The color codes indicate how likely one of the next tones is. Thus in the high regularity condition (lower panel), according to the transition matrix, tone 4 always follows tone 1. However, this is not correct given the triplet representation on the left. The highest tone (932Hz) is always followed by the second highest tone in the tone frequency representation. Therefore, the yellow field (100%) should be at matrix position (1,2). The figure only makes sense if the bottom row refers to the highest tone and the top row to the lowest tone. I also assume that the transition matrices show the statistics across the entire experiment given that in the upper panel transition probabilities in the triplet representation do not match transition probabilities in the matrix. For example, I count 9 tones with a frequency of 262 Hz. Four out of these nine tones are followed by tone 2. This results in a probability of 44%. But the color code suggests that 25% is the maximal transition probability.

We thank the reviewer for spotting this. Indeed, in the original manuscript the top row corresponded to the lowest tone (262 Hz). We agree that this was misleading. Thus, we followed the reviewer’s suggestions and flipped the transition matrix. Concerning the statistics, as the reviewer correctly assumes, the left panel only shows a brief section of the full sequences. Thus, the statistics in these subsections do not match the transition matrices, which show the statistics across the full sequences.

At some points, I found the writing unnecessarily complicated. For example, "the betweenness centrality of synergy and redundancy revealed the frontoparietal and temporal cortices as hubs for both information components."

We followed the reviewers suggestion and simplified this sentence. It now reads: “[...] Also, betweenness centrality, a measure of node importance, revealed frontoparietal and temporal cortices as hubs [...]” (line 208)

Then the authors write (line 234) that this paradigm represents a naturalistic presentation. The paradigm is indeed less artificial than an oddball paradigm, but it is still far from a naturalistic setting.

We agree with the reviewer and only meant ‘naturalistic’ in a comparative sense. To avoid any potential misunderstanding, we removed all uses of the term except for one case where we explicitly use it in a comparative sense: “[...] This computational approach allowed us to study prediction error encoding beyond the traditional comparison of standard and deviant stimuli in trial-based oddball paradigms (10, 11, 15, 49) in a continuous, and thus more naturalistic, sequence paradigm (67). [...]” (line 257).

By addressing these points, the manuscript will become clearer and more accessible to readers, enhancing the impact and understanding of this important work.

Again, we thank the reviewer for the positive assessment of our work and for the very helpful comments and suggestions.